# Rapid Morphological Measurement Method of Aortic Dissection Stent Based on Spatial Observation Point Set

**DOI:** 10.3390/bioengineering10020139

**Published:** 2023-01-20

**Authors:** Mateng Bai, Da Li, Kaiyao Xu, Shuyu Ouyang, Ding Yuan, Tinghui Zheng

**Affiliations:** 1Department of Applied Mechanics, Sichuan University, No. 24 South Section 1, Chengdu 610065, China; 2Yibin Institute of Industrial Technology, Sichuan University Yibin Park, Yibin 644600, China; 3Department of Biomedical Engineering, Sichuan University, Chengdu 610065, China; 4Division of Vascular Surgery, Department of General Surgery, West China Hospital, Sichuan University, No. 37 Guo Xue Xiang, Chengdu 610041, China; 5Med-X Center for Informatics, Sichuan University, Chengdu 610041, China

**Keywords:** aortic dissection, thoracic aortic stent, completeness dataset, statistical time complexity, morphological parameters

## Abstract

Objectives: Post-operative stent morphology of aortic dissection patients is important for performing clinical diagnosis and prognostic assessment. However, stent morphologies still need to be manually measured, which is a process prone to errors, high time consumption and difficulty in exploiting inter-data associations. Herein, we propose a method based on the stepwise combination of basic, non-divisible data sets to quickly obtain morphological parameters with high accuracy. Methods: We performed the 3D reconstruction of 109 post-operative follow-up CT image data from 26 patients using mimics software. By extracting the spatial locations of the basic morphological observation points on the stent, we defined a basic and non-reducible set of observation points. Further, we implemented a fully automatic stent segmentation and an observation point extraction algorithm. We analyzed the stability and accuracy of the algorithms on a test set containing 8 cases and 408 points. Based on this dataset, we calculated three morphological parameters of different complexity for the different spatial structural features exhibited by the stent. Finally, we compared the two measurement schemes in four aspects: data variability, data stability, statistical process complexity and algorithmic error. Results: The statistical results of the two methods on two low-complexity morphological parameters (spatial position of stent end and vascular stent end-slip volume) show good agreement (*n* = 26, *P*_1_, *P*_2_ < 0.001, *r*_1_ = 0.992, *r*_2_ = 0.988). The statistics of the proposed method for the morphological parameters of medium complexity (proximal support ring feature diameter and distal support ring feature diameter) avoid the errors caused by manual extraction, and the magnitude of this correction to the traditional method does not exceed 4 mm with an average correction of 1.38 mm. Meanwhile, our proposed automatic observation point extraction method has only 2.2% error rate on the test set, and the average spatial distance from the manually marked observation points is 0.73 mm. Thus, the proposed method is able to rapidly and accurately measure the stent circumferential deflection angle, which is highly complex and cannot be measured using traditional methods. Conclusions: The proposed method can significantly reduce the statistical observation time and information processing cost compared to the traditional morphological observation methods. Moreover, when new morphological parameters are required, one can quickly and accurately obtain the target parameters by new “combinatorial functions.” Iterative modification of the data set itself is avoided.

## 1. Introduction

Type B aortic dissection(TBAD) refers to the tearing of the aorta intima beyond the aortic arch; the blood enters the middle layer of the aorta, develops along the longitudinal axis of the aorta and divides the aorta into a true lumen and a false lumen [1,2]. Thoracic endovascular aortic repair (TEVAR) is characterized by implanting a stent in the thoracic aorta to cover the proximal tear and expand the true lumen, thereby restoring the blood supply to the aorta; it is currently the most widely adopted treatment for TBAD [3,4]. Compared with conventional open surgical treatment, TEVAR has the advantages of less trauma, shorter operation time and less blood transfusion [5,6,7]. There have been many serious complications, especially the formation of distal new entry tears, aneurysms or pseudo aneurysms induced by stent implantation—having mortality rates as high as 25%—which greatly affect the mid- and long-term outcomes of TEVAR.

Evaluating the morphological changes of the implanted stents helps to assess the treatment outcome and risk of post-operative complications [8,9,10]. The stent faces a complicated mechanical environment after it is implanted in the human body [11,12], including the impact force by intravascular blood flow and compression pressure by the remained false lumen. In such a complex environment, the stent is highly likely to move and deform; this is the main cause to the formation of bird beak configuration, distal reentry tears, non-thrombosis false lumen and leakage following TEVAR [13,14,15,16]. For example, Sophocleous et al. [17] found that aortic arch presenting a more Gothic arch architecture was associated with reduced ejection fraction, increased end-diastolic volume and ventricular mass. Sun et al. [18] found that more re-entry tears and the primary tear proximal to the arch were associated with a higher risk of negative remodeling after TEVAR based on the post-TEVAR variations of the false lumen volume. In addition, Li et al. [19] proposed the use of the degree of question mark to define the aortic angulation and revealed that the greater the question mark degree was, the less likely it was to form a complete thrombus of the post-operative false lumen.

However, up to now, these stent-related morphological parameters following TEVAR are all manually measured either on CTA images or reconstructed 3D models based on CTA images, which are not only time-consuming but error-prone. In addition, the measurement results may vary among individuals and the data accuracy decreases rapidly with increasing statistical size [18,20]. Rychla et al. [21] compared three different parameter measurement methods and stent selection schemes, and found significant differences in the parameter results obtained by the different “measurement methods” and “proposed schemes”. In addition, the manual measured morphological parameters are very limited and generally can only provide simplified stent information such as diameters, length and angles. It is impossible to quantitatively characterize the overall spatial torsion of the stent and important deformation features of stents [22,23].

Actually, the stent morphology is defined by the spatial coordinates of the stent points, and its morphological changes are caused by changes in their spatial coordinates. If a set of points can be determined to represent the overall morphology of the stent, any stent morphological parameter can be obtained using the spatial coordinates of these points. Accordingly, in the current study, we proposed a computer-based calculation of stent morphological parameters. We first selected the feature observation points (representative points) on the stent and determined their spatial coordinates and completed the calculation of the stent morphological variables by defined mathematical functions. The core of the proposed method is the definition and analysis of observation points. We can directly construct them from spatial observation points when new morphological parameters are needed, rather than re-count the data samples, which not only avoids the errors by manual measurements, but also obtains high-complexity morphological parameters rapidly and accurately, thereby providing important post-TEVAR prognostic information for better evaluation of treatment outcomes. Furthermore, we designed fully automatic segmentation to the stent and extraction methods to the observation point. In summary, we proposed and built a complete stent morphological analysis process which includes the automatic stent segmentation method, the automatic observation point extraction method, and the parameter combination system mentioned above. We refer to this process as the rapid morphological measurement method (RMMM).

## 2. Methods

This study was conducted in accordance with the principles of the Declaration of Helsinki and met the requirements of the medical ethics. The Ethical Review Committee of the West China Hospital of Sichuan University (Chengdu, Sichuan, China) approved this research. Patient approval and informed consent were waived because the study was purely observational, retrospective in nature and used anonymized data.

The detailed CTA images of all patients with TBAD who underwent TEVAR at the West China Hospital, Sichuan University, between December 2011 and October 2018 were collected. We excluded non-imagable CTA [24,25] because, under the influence of the nonuniformly distributed contrast media, the stents become blurred and incomplete. Cases with missing patient information data were also excluded because these could not be classified accurately and the follow-up time could not be calculated. Finally, this study included 26 patients (Appendix A Table A1) with a 3-year follow-up period (Appendix A Figure A1).

The proposed method rapidly generates morphological parameters by counting the spatial location of the stent points. This replaces the part of the traditional method that analyzes the parameters case by case. The differences of the two methods are illustrated in Figure 1, and the specific steps are as follows.

### 2.1. Definition and Extraction of Basic Observation Points of Vascular Stents

First, the vascular stents were manually reconstructed by Mimics (V21.0, Materialise, Plymouth, MI, USA) based on patient-specific CTA images (Figure 2). Since the HU value of the vascular region was between 300 and 700 in our CT dataset, in the process of stent reconstruction, we set the lowest threshold of 1200 to reduce the probability of high HU value regions in the vessel being identified as the stent, and the highest threshold of 2200 to reduce the influence of stent artifacts. After coarse segmentation using that threshold, we selected the stent part according to the regional connectivity through the region grow function.

Second, we extracted the observation points on the stent manually. The observation points were extracted by two doctors, respectively. When their judgments on a point differed by more than 1.5 mm, the third experienced clinician made the final verdict. The observation points are determined on the stent. The peak points of each stent ring are used as observation points (Appendix A Figure A2), which are used to form the basic elements describing the spatial morphology of the stent. The peak point was selected as the observation point because the peak point is easy to select and its location is accurate. Further, we must select the skeletal localization point on the CT image as the origin of the spatial coordinate system to register different spatial coordinates. Therefore, we choose the origin point on the 12th bone upwards, starting from the caudal spine, which is constrained in three degrees of freedom XYZ by the morphological features of this bone (Appendix A Figure A3).

Third, we determine the midpoint of the support ring. For a support ring containing *N* peak points, the point-mid of the ring is defined as the average of the spatial coordinates of all peak points of the ring, and its *x*, *y* and *z* coordinates are calculated in Equation (1):(1)$$\{\begin{array}{l} \;point\_mid{\;}_{x}=\frac{{\sum^{\;}}_{i=1}^{N}P_{ix}}{N}\\ \;point\_mid{\;}_{y}=\frac{{\sum^{\;}}_{i=1}^{N}P_{iy}}{N}\\ \;point\_mid{\;}_{z}=\frac{{\sum^{\;}}_{i=1}^{N}P_{iz}}{N} \end{array}$$
where *point_mid_x_*, *point_mid_y_* and *point_mid_z_* are the *x*, *y* and *z* components of the point coordinates on the *X*, *Y* and *Z* axes of the stent ring, respectively. *Pi* denotes the coordinates of the peak point on the stent ring and *N* is the number of peak points on the stent ring.

Finally, we perform the visualization of stent parameters. The Matplotlib library [26] was used to visualize the observation points, followed by synthesis and validation of the complete follow-up data (Figure 3).

### 2.2. Automatic Stent Segmentation and Observation Point Extraction Method

Manual segmentation of the stent and extraction of observation points will consume a lot of time, so we proposed a fully automatic segmentation method using a deep learning network based on VNet [27]. In addition, we designed a fully automatic algorithm to extract the observation points based on the segmentation model replacing the manual observation point extraction process. The detailed steps are as follows:

Firstly, we built the VNet model [27] to complete the automatic segmentation task of stent. We randomly selected 50 out of 58 stent segmentation masks to train the network. Two doctors labeled the scans separately. The dice_loss [28] coefficient was used to measure the difference between the two annotations. When dice_loss exceeds 0.05, the third experienced doctor will make the final decision. Before training, all CT files were linearly interpolated into a spatial resolution of 1 mm × 1 mm × 1 mm, then we crop the space to 96 mm × 144 mm × 160 mm. The Adam optimizer was selected for parameter update. The dice_loss was used to monitor and evaluate the prediction performance of the network.

After that, we used the connectivity function in the ITK library [29,30] to check the volume of each connected region in the prediction result. In this way, we can obtain the segmentation model of the separation of each support ring.

Next, we calculated the center of gravity of each ring and obtained the center line of the stent by the cubic spline interpolation. The peak point on the stent was calculated through the center line and the stent voxel model. The pseudo-code and schematic diagram of the algorithm are shown in Figure 4.

After performing the algorithm described in Figure 4b, A, B and G were eliminated and C, D and E were retained.

Finally, each retained peak point was expanded into a sphere with a radius of 2 mm. In this manner, sufficiently close peak points became connected to each other. The final output was the center of gravity of each connected region. With the clustering algorithm, two points C and D in Figure 4a were correctly identified as a peak point, which was the midpoint of line segment CD.

### 2.3. Representative Morphological Parameters of Stent

After completing the statistics and normalization of the observation points of the stent, we proposed and analyzed the following three morphological parameters with different complexity. These parameters were fully calculated automatically with the use of basic data:

#### 2.3.1. Stent End-Slip Vector

The stent end-slip vector describes the spatial motion of the end of the stent. As shown in Figure 5, T1, T2 and T3 are the models obtained by CT extraction for the three post-operative follow-up modeling processes, the coordinates of P_O1_, P_O2_ and P_O3_ and the midpoint of the stent end-loop ring were obtained for the three periods T1, T2 and T3, respectively. Subsequently, according to Equation (2), we can calculate the end-slip of the stent between periods *i* and *j*:(2)$$\{\begin{array}{l} \Delta x_{ij}=Po_{i\_}x-Po_{j}\_x\\ \Delta y_{ij}=Po_{i\_}y-Po_{j\_}y\\ \Delta z_{ij}=Po_{i\_}z-Po_{j\_}z \end{array}$$
where Δ*x*, Δ*y* and Δ*z* represent the components of the end-of-frame slip on *x*, *y* and *z*, respectively. *i* and *j* footnotes indicate that the slip occurs between periods *i* and *j*.

#### 2.3.2. Radial Characteristic Diameter Change of Support Ring

Because of the large variability of the support ring shape in the human body, most of which cannot be approximated as a positive polygon for analysis, we defined the characteristic length *ρ* of the support ring as the maximum value of the distance between the peak points of the stent (Figure 6). Subsequently, the ratio of characteristic lengths for each period of the same stent ring was used to reflect the magnitude of the radial relative deformation rate of the stent. Their relationship is defined in Equation (3).
$$\rho =\mathrm{MAX}\left(\left|P_{i}P_{j}\right|\right)$$
(3)$$R_{k}={\left(\frac{\rho_{k}}{\rho_{1}}\right)}^{2}$$

(1 ≤ *i*, *j* ≤ number of peak stent ring points).

*ρ_k_* is the characteristic length of the support ring in period k, *P_i_P_j_* is the line segment consisting of two characteristic observation points on the bracket ring in that period and *R*_k_ is the “ring rate” in period k of that ring.

#### 2.3.3. Stent Ring Deflection Angle

Figure 7 shows the sequentially numbered stent rings from 1 to M from the proximal to the distal end. The coordinates of the relative positions of the M stent ring point_mid are calculated. The data of the stent ring numbered k(1 ≤ k ≤ M) in the two follow-up periods of T1 (Figure 7a(T1)) and T2 (Figure 7a(T2)) were taken as an example. First, the vector map (Figure 7b) was obtained by overlapping the calibration of the point_mid of the two periods. Notably, the five points of ABCDE are not necessarily strictly in the same spatial plane *β* in the actual calculation process. We used the vector O_k+1_O_k_ consisting of the midpoints of the stent ring numbered k and k + 1 as an approximation to the vertical plane vector of the spatial plane *β*. The magnitude of the projection angle of ∠ P_j_OP_j_′ on the plane *β* is calculated in the vector diagram as the magnitude of the deflection angle of point *P_j_* (1 ≤ *j* ≤ N), and the average value of the deflection angle of all peak points of this support ring is taken as the magnitude of the deflection angle of the ring (Figure 7c). Their definitions and relationships are defined in Equation (4).
(4)$$\alpha_{j}=\arcsin\left(\frac{\vec{OP_{j}}\times \vec{OP_{j}^{\prime }}}{\left|OP_{j}\right|\left|OP_{j}^{\prime }\right|}\cdot \frac{\vec{O_{i+1}O_{i}}}{\left|O_{i+1}O_{i}\right|}\right)\;\;\;\;\;\;\;\beta =\frac{{\sum^{\;}}_{j=1}^{N}\alpha_{j}}{N}$$

Here, *N* is the number of peak points of the stent ring; *O_i_* is the ring center point of the ring numbered *i*; *P_j_* is a peak point on this support ring; *α_j_* is the magnitude of the deflection angle corresponding to point *P_j_*, and *β* is the magnitude of the deflection angle corresponding to the ring.

## 3. Results

### 3.1. Automatic Stent Segmentation and Observation Point Extraction

We trained the VNet network using the method described in Section 2.2. The network reached stable after 400 epochs. On the test set, we calculated the dice_loss between the network prediction result and the manually segmented annotation. The dice_loss of the eight models in the test set was all less than 0.075. Further, we tested the reliability of the automatic observation point extraction method. We performed both manual method and fully automatic method on the test set. The spatial distance D was used to show the difference between the observations points extracted by the different methods. Table 1 lists the performance of fully automatic observation point extraction methods on the test set. The visualization results of the fully automatic segmentation extraction method are shown in Figure 8.

### 3.2. Vascular Stent End-Slip Volume

Table 2 shows that the measurements of the two methods on the spatial position of the stent ring, and the amount of slip at the end of the stent are in excellent agreement. The difference between the two methods for the measurement of the Z-directional component of the stent end does not exceed 3 mm, and the difference between the measurement of the spatial position of each support ring does not exceed 2.5 mm. The relative errors of the two methods (0.18%, 11.25%) are below 15%, indicating that the proposed method has good stability and reliability for calculating parameters with low complexity and geometrically defined consistent features (position parameters and vector parameters) (Figure 9).

### 3.3. Radial Characteristic Diameter Change of Support Ring

Table 3 shows the variability of the two methods in terms of parameter statistics of medium complexity. The correction of the measurement of the “proximal support ring feature diameter” by the proposed method does not exceed 4 mm, and the average correction is 1.14 mm. The correction of the measurement of the “distal support ring feature diameter” does not exceed 5 mm, and the average correction is 1.38 mm (Figure 10).

### 3.4. Stent Ring Deflection Angle

Using the stent deflection angle definition in 2.3c, the spatial distribution analysis and temporal correlation analysis of the circumferential deflection angle for each case were performed using the stent observation point set and the corresponding three-dimensional visualization post-processing (Figure 11).

## 4. Discussion

TEVAR is gradually becoming the main response option for thoracic aortic dissection [31,32,33] due to its minimally invasive and safe nature [34,35,36]. However, in the complex environment of the human body, vascular stents are prone to deformation, at the same time, this deformation will cause a variety of complications [15]. Previous studies have shown the clinical importance of postoperative morphological measurements and its ability to provide valid prognostic information [37,38]. At the same time, morphological changes in the stent are closely related to various acute complications [39]. Therefore, the use of morphological parameter analysis to assess stent-vessel space deformation after TEVAR plays an important role in reducing the risk of death.

At the same time, previous studies have shown that different methods of parameter extraction, hemodynamic environments [40,41,42] and changes in physiological conditions [43] can lead to widely varying results. Furthermore, different measurement methods combined with different case series [44,45,46,47] and medical recommendations [9,48] can give very different results. As a matter of fact, the consistency and accuracy of morphological parameters obtained by manual measurements have been questioned [49,50,51]. Therefore, we need a strict definition to regulate morphological parameters of different complexity; while ensuring conciseness and accuracy, they can be quickly observed and analyzed.

To date, morphological parameters are still measured manually by clinicians, a process which is error-prone, and measurement results may vary among individuals. Moreover, counting morphological parameters one by one leads to excessive time consumption, and the data accuracy decreases rapidly with increasing statistical size. Furthermore, the manual measurement makes it impossible to quantitatively characterize some important deformation features of stents, such as the stent ring deflection. At the same time, traditional morphological measurement methods focus on the direct extraction of data and simple combinations [52,53]. Therefore, they face problems of large errors, high time complexity and low efficiency when the data sample size is large. This study proposed a fully automatic method of feature observation point (representative point) selection on the stent and completed the calculation from low-complexity parameters (spatial coordinates of stent points) to high-complexity parameters (stent morphological variables) by mathematical functions, thus obtaining various morphological parameters of the stent rapidly and accurately. Computer programming replaces the human effort to quickly and accurately combine low-complexity basic data into high-complexity parameters step by step and perform the statistical analysis and post-processing. In this manner, it has significant advantages in terms of time consumption and accuracy and provides morphological parameters with high accuracy and good stability for clinical experimental analysis after TEVAR for aortic type B dissection.

By comparing different morphological parameter extraction processes, we explored the performance of the two methods on parameter extraction tasks of different complexity. The analysis of three important morphological characteristics of the stent after TEVAR for aortic type B dissection showed that:(1)The proposed method can accomplish accurate statistics for low-complexity parameters within a shorter time. The stent slip space vector obtained by the proposed method (with the lowest number of participation points *n* = 2 and the lowest measurement complexity) is in excellent agreement with the traditional method. This verifies the accuracy and stability of the “combined” measurement method on parameters with low combined complexity. Further, the proposed method eliminates the need to repeat the statistics for basic parameters (spatial basis points), thus significantly reducing the statistical time needed for the parameters;(2)The proposed method can effectively correct “manual measurement errors” in parameters of medium complexity. For example, the radial characteristic diameter of the stent involves a comparative analysis of the length of multiple points on the stent ring. The increase in the amount of data involved in the operation (the number of participating points *n* = 16, which has medium measurement complexity) causes an increase in data complexity and the gradual appearance of accumulation of manual measurement errors at each observation point, which causes the differences between the traditional measurement and the proposed method (Appendix A Figure A4). Furthermore, the proposed method not only avoids the risk of statistical errors by using the basic data points for the “combination operation,” but can also be programmed to incorporate more “combination functions” into the operating system, thus exhibiting higher data accuracy and measurement speed. This minimizes the cost of human statistical analysis;(3)The proposed method allows fast and accurate measurement of morphological parameters of high complexity, for which it is difficult to make statistics by traditional methods. For example, we define the stent circumferential deflection angle (with high combinatorial complexity and *n* = 26 points involved in the combination). Therefore, the complexity of the calculation method of the stent torsional deflection angle composition by far exceeds the reasonable measurement statistics acceptable by the traditional method, and the statistical analysis of the composition of this quantity one by one is extremely time-consuming. The proposed method, however, does not need to expand the base data set owing to its “combinatorial” nature, and only expands the set of generalized combinatorial functions to rapidly perform combinatorial analysis and visualization of the quantity using the computer. This demonstrates the good scalability and speed advantages of the proposed method for the calculation of parameters with high combinatorial complexity.

This study currently suffers from the following limitations:

### 4.1. Improved Image Registration at Different Follow-Up Periods

In this study, we used marker points on bones as the spatial origin of the CT model for each follow-up period for vector calibration, ignoring the effect caused by the small deflection angle of the patient’s body in different follow-up periods. In subsequent studies, attempts can be made to supplement the correction of errors caused by spatial angle rotation by introducing other body positioning points at appropriate distances [28,54]. In future studies, we hope to introduce non-rigid registration [55] into this work. Non-rigid registration [56] can better reflect the local enlargement and contraction of the stent and other more complex deformation. This requires us to select more mark points (such as the vertex or center on the manubrium sternum) for registration. At the same time, we note that abandoning the mark point and using image grayscale information for registration is also an interesting direction. Furthermore, the definition of “deformation field” in non-rigid registration is worthy of exploring. The “observation point” is a discrete description. In the following study, we hope to refine the rapid morphological measurement method by establishing a continuous form through non-rigid registration.

### 4.2. Method Efficiency Is Influenced by Sample Size

The scale of operations using the proposed method is of concern. If the target extracted morphological feature volume is of low complexity, and the number of models is small, the disadvantage in the cost of time incurred when spending a large upfront cost to build the base data set may far outweigh the small accuracy advantage it exhibits on low-complexity data. At this point, using traditional methods to directly model statistics on low-complexity data is a better choice. Further, if a more robust automatic extraction method of basic observations can be proposed using deep learning [57,58,59] or spatial threshold partitioning algorithms, the efficiency of the atomic dataset system proposed in this study can be further improved in terms of the extraction of raw data. This would render a complete set of morphological observation extraction analysis systems, suitable for each complex combination. In future work, we expect to use the graph network [60,61] to analyze unstructured data. This will further broaden the scope of the “combination algorithm” and help us better understand the basic pattern of the data.

### 4.3. Integrity of the Original Information

The method proposed in this paper uses spatial points as the most basic data. The spatial points are sufficient to combine all parameters related to the morphology of the vascular stent. Further, can smaller, more comprehensive basic data units be used? In the next study, we expect to be able to use grayscale features as a more basic data unit. In this way, the hemodynamic parameters can be linked to the basic data unit. Thus, a more comprehensive data model can be built. Further, we can combine more observation techniques. Compared with CT, MRI (Magnetic resonance imaging) can capture the shape boundary more clearly which helps us to get more accurate original data. However, the resolution of MRI is generally lower than CT, which leads to the degradation of segmentation quality and statistical accuracy [62,63,64]. In addition, 4D flow MRI can capture flow field structures [65]. In future work, we hope to couple the flow field and the “observation point”. By analyzing the changes in flow field and stent data during each follow-up period, we will be able to elucidate the major and minor factors causing such changes. This will help us to better understand the action mechanism of different flow states such as “vortex” and “swirling flow”. It is possible to analyze the relationship between hemodynamic parameters and morphological changes with this highly accurate analysis method.

### 4.4. Clinical Outcome and Prognostic

It is important to link comprehensive and accurate measurements for clinical outcomes for patients [66,67]. In future studies, we hope to establish a stable mathematical prediction model. The model relies on accurate and comprehensive statistics of patient information. Using the prediction model, we can accurately predict the deformation and displacement trend of the stent. Further, through data combination and reconstruction, we hope to establish the relationship between data and clinical outcomes. It is of great clinical value to use PCA (principal component analysis) [68] to find the parameters that have the main impact on clinical outcomes [69].

## 5. Conclusions

We define a complete computational analysis based on stent observation points and the generalized “combinatorial function” and propose its fully automatic implementation. Experimental operations and analyses are performed on three morphological parameters of different stent complexity. Based on this, we compared the two methods in terms of analysis time complexity, accuracy and scalability. We found that the morphological statistical method based on the underlying data set not only reliably analyzes morphological parameters of different complexity, but also holds great advantages in terms of time complexity, scalability, statistical accuracy and data visualization capability. The advantages of the proposed method in terms of speed, accuracy and scalability not only provide measurement support for morphological observations of various degrees of complexity, but also make it possible to build coupled stent-vessel models [70] based on large-scale datasets.

## Figures and Tables

**Figure 1 bioengineering-10-00139-f001:**
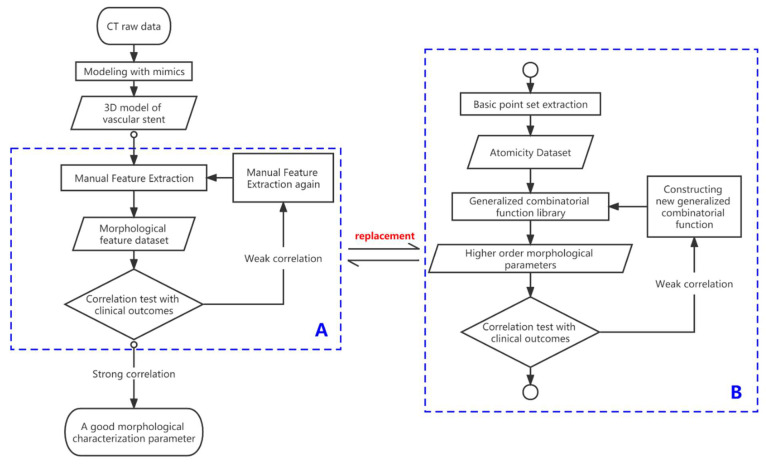
Block diagram of traditional and novel processes for measuring morphological parameters. The left half of the image shows the process of analyzing morphological parameters using traditional method. The novel method in this section shown in dashed box (**B**) replaces the traditional process shown in dashed box (**A**).

**Figure 2 bioengineering-10-00139-f002:**
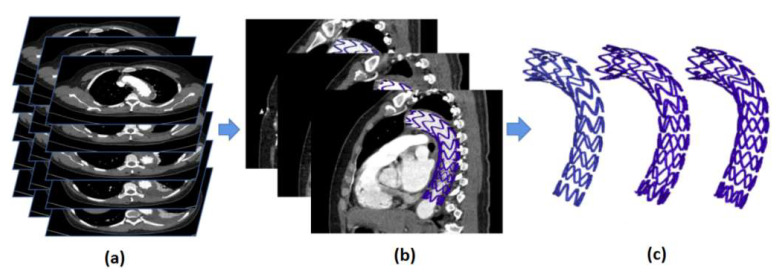
Schematic diagram of vascular stent reconstruction by CT. (**a**) CT patient data; (**b**) position of reconstructed stent model in the CT; (**c**) spatial shape of reconstructed stent.

**Figure 3 bioengineering-10-00139-f003:**
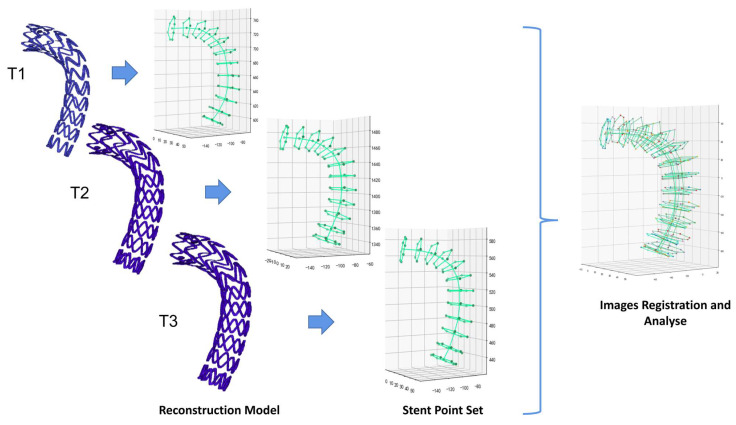
Visualization and registration of stent points. (T1, T2 and T3 denote models at 3, 6 and 12 months, respectively).

**Figure 4 bioengineering-10-00139-f004:**
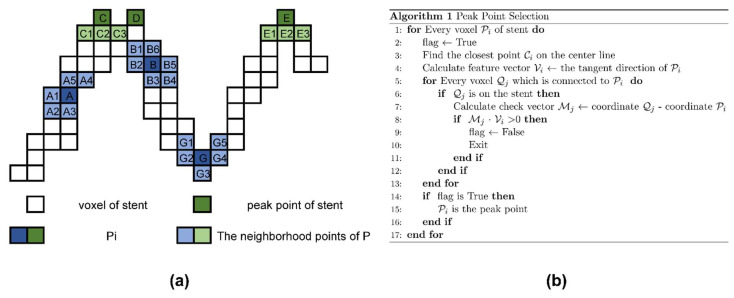
Pseudocode and schematic diagram of peak point extraction algorithm: (**a**) two-dimensional form of observation point extraction algorithm; (**b**) pseudocode for peak point calculation.

**Figure 5 bioengineering-10-00139-f005:**
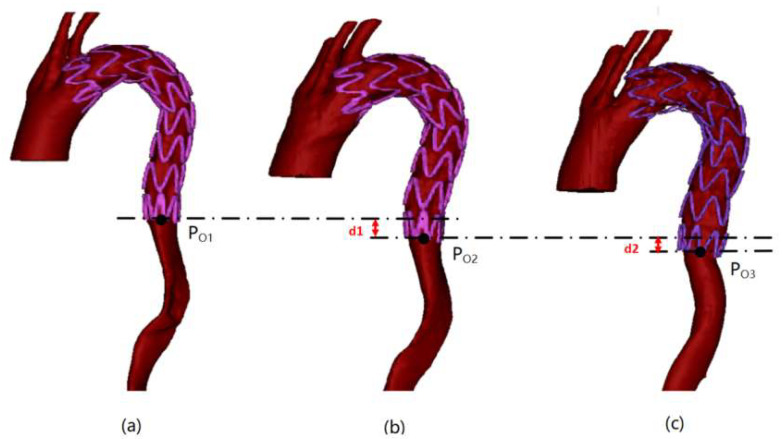
Schematic diagram of stent end-slip vector: (**a**–**c**) are the results of the 3, 6 and 12 months follow-ups, respectively.

**Figure 6 bioengineering-10-00139-f006:**
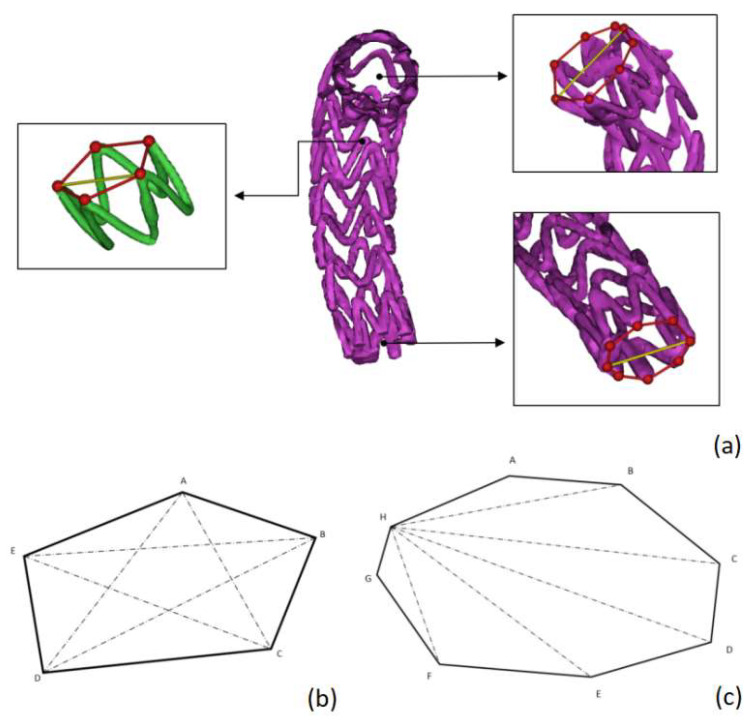
Schematic diagram of stent characteristic diameter: (**a**) the red line marks the shape of the outer connection of each ring of the stent, and the yellow line marks the selection point of the characteristic diameter of the ring; (**b**) the line of the selection of the characteristic diameter of the five-point ring; (**c**) the line of the selection of the characteristic diameter of the eight-point ring during the traversal of the H-point.

**Figure 7 bioengineering-10-00139-f007:**
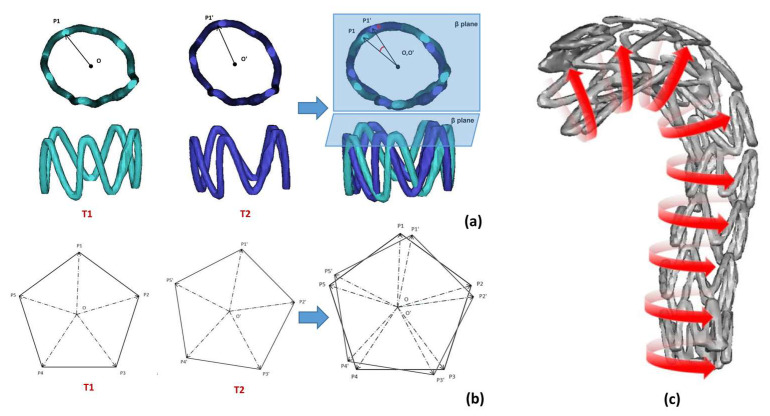
Schematic diagram of deflection angle of support ring: (**a**) stent ring deflection angle definition during two different follow-ups of the same patient; (**b**) geometric definition of deflection angle of stent ring in the top view; (**c**) visual diagram of stent ring deflection angle.

**Figure 8 bioengineering-10-00139-f008:**
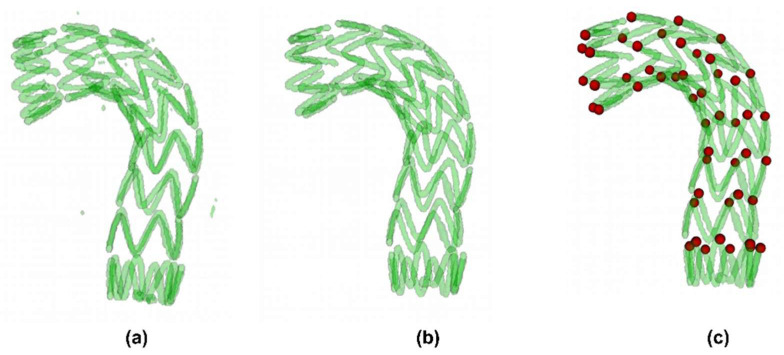
Visualization of the fully automatic segmentation extraction method: (**a**) automatic segmentation results by VNet; (**b**) continuous region analysis results; (**c**) output of observation extraction algorithm.

**Figure 9 bioengineering-10-00139-f009:**
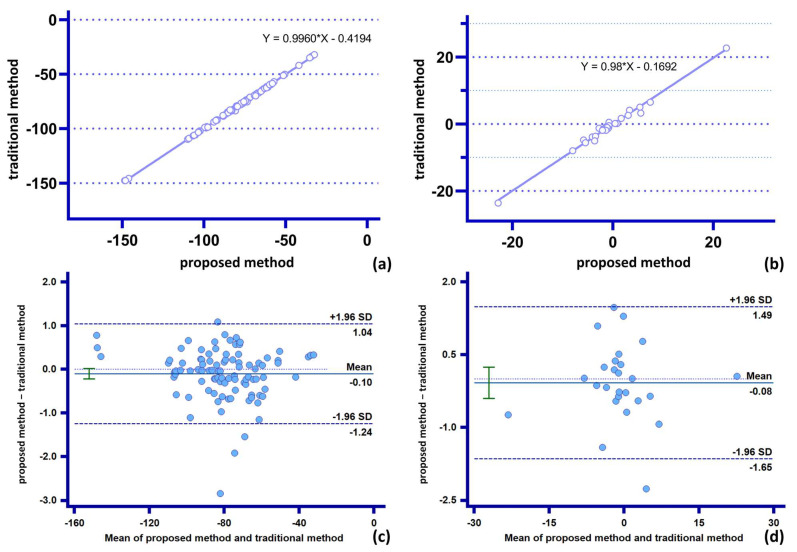
Comparison of consistency of two methods for low-complexity parameter statistics: (**a**,**c**) measurement and analysis of relative spatial position of stent end by the two methods; (**b**,**d**) measurement and analysis of Z-directional component of stent slip by the two methods.

**Figure 10 bioengineering-10-00139-f010:**
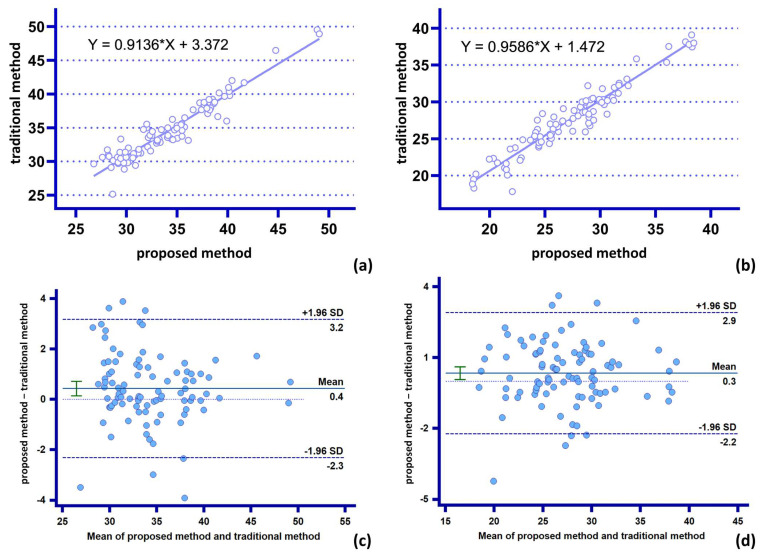
Comparison of agreement between the two methods for the statistics of parameters of moderate complexity: (**a**,**c**) measurement and analysis of proximal characteristic diameter of stent for both methods; (**b**,**d**) measurement and analysis of distal characteristic diameter of stent for both methods.

**Figure 11 bioengineering-10-00139-f011:**
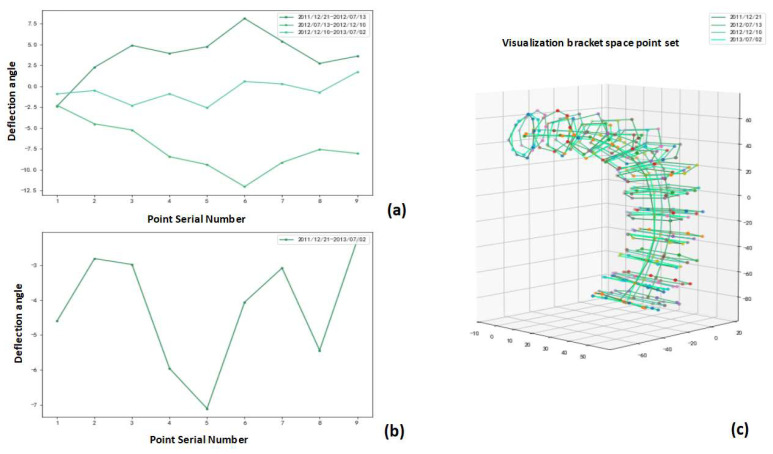
Analysis of stent ring deflection angles output by proposed method: (**a**) distribution of stent ring deflection angles for same patient during four follow-up visits; (**b**) deflection angles of each stent ring for a patient during the first and last follow-up visits; (**c**) four periods normalized for patient.

**Table 1 bioengineering-10-00139-t001:** Results of the fully automatic segmentation extraction method on test set.

Number of Checkpoints	Missing Extraction	D_Mean (mm)	D_Max (mm)
408	9 (2.2%)	0.73	3.55

**Table 2 bioengineering-10-00139-t002:** Statistical comparison of two methods in spatial position of stent end with Z-directional component of slip.

Variable	Difference Mean (mm)	Difference SD	R2
Spatial position of stent end	0.1576	0.58	0.9992
Vascular stent end-slip volume	0.0997	0.81	0.98849

**Table 3 bioengineering-10-00139-t003:** Statistical comparison of two methods on measurement of characteristic diameters.

Variable	Mean (mm)	SD	R2
Proximal support ring feature diameter	0.337	1.35	0.9022
Distal support ring feature diameter	0.613	1.30	0.9204

## Data Availability

The data presented in this study are available on request from the corresponding author. The data are not publicly available due to ethical restrictions.

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
