# Peer review of "Rapid Morphological Measurement Method of Aortic Dissection Stent Based on Spatial Observation Point Set"

_bioengineering, 2023, doi:10.3390/bioengineering10020139_

Round 1

Reviewer 1 Report (Previous Reviewer 1)

All comments and suggestions were properly addressed

Author Response

Thank you for your suggestion!

Reviewer 2 Report (Previous Reviewer 3)

The authors have appropriately addressed all of the concerns that I raised in my first review. I have no further comments.

Author Response

Thank you for your suggestion!

Reviewer 3 Report (Previous Reviewer 4)

The manuscript is improved.  However, there are still some editing issues the authors should addressed.  The following ae suggestions/comments regarding those issues.  Line 69, "...found that in the larger tear areas, more ...".  Line 96, "...which not only avoids the errors ...".  Line 100, "...extraction methods to the segment of the stent and extract ...".  Line 117, "...Table 1) with a 5-year follow-up period (Appendix Fig. 1)."  Line 133, "...coarse segmentation using that threshold, we ...".  Lines 134 & 135, "...the region grow function."  Line 176, "interpolated into a spatial resolution of ...".  Line 202, "... parameters were fully calculated automatically ...".  Line 418, "...complex combination.  In future work, we expect ...".  Line 433, "field structures [65]."  Line 441, "...accurate measurements for clinical ...".  Lines 441 & 442, "...clinical outcomes in patients [66,67].  In future studies, we hope to ...".

Author Response

This manuscript is a resubmission of an earlier submission. The following is a list of the peer review reports and author responses from that submission.

Round 1

Reviewer 1 Report

A quantitative assesment of the endograft malapositioning is carried out in this study to propose a new approach for evaluating the likelihood of the device failure. The paper needs a revision of English with several comments to be addressed prior to publication.

page 2 line 61. among the cited paper, please consider to cite the following on in-vitro analisis of the bird-beak configuration

Pasta S, Scardulla F, Rinaudo A, Raffa GM, D'Ancona G, Pilato M, Scardulla C. An In Vitro Phantom Study on the Role of the Bird-Beak Configuration in Endograft Infolding in the Aortic Arch. J Endovasc Ther. 2016 Feb;23(1):172-81. doi: 10.1177/1526602815611888. Epub 2015 Oct 23. PMID: 26496956.

page 2 line 64. please remove "And" at the begining of the sentence and add the reference just after the author name.

page e line 77: "in addition, more importantly", please remove one of these term. Overall, the paper need to be revised for the english.

page 3 Figure 1: please add letter to describe the traditional and novel measurement approaches

page 3 line 102: is the segmetnation done automatically or by semi-automatic segmentation? which threshold value have you used? please add more details as this part may influence all the work here proposed.

page 4 line 123. it is not clear if this measurments are automatically done or performed manually. please specify these aspects. Moreover, are these measurments done only by one user. have you tested inter and intra-observer variability?

page 7 line 14: are these measurements done automatically? please specify this aspects.

page 7 Figure 7 usually a balnd-altman plot is more indicated for comparing the measurements of two approaches.

page 8 Figure 8 please consider bland-altman comparison and fix the legend that is wrong for both x and y axes.

page 8, line 232. please consider to cite the following paper on the paremetrization of ascending aort to chracterize the haemodynamic.

Pasta S, Gentile G, Raffa GM, Scardulla F, Bellavia D, Luca A, Pilato M, Scardulla C. Three-dimensional parametric modeling of bicuspid aortopathy and comparison with computational flow predictions. Artif Organs. 2017 Sep;41(9):E92-E102. doi: 10.1111/aor.12866. Epub 2017 Feb 10. PMID: 28185277.

Reviewer 2 Report

The abstract is well structured and introduces reader adequately to the scope and the findings of the article as well as its clinical implications.

In general it is a well written, structured, concrete and interesting article.

The introduction is well referenced and supports the aim of the study. The scope of the study is clearly stated. However, a more detailed reference regarding imaging findings that are predictive of positive or negative remodeling in patients who have undergone TEVAR, I believe that it would very helpful for the readers.

The methods explain in detail the processes for measuring the morphological parametres of the stent and the time periods for the follow up measurements. The figures in the methods are very helpful. There is no reference regarding inclusion and exclusion criteria regarding the patients selection. Moreover there is no reference to the approval of the study from the Hospital's or other Bioethics Committee.

The results I believe that need to be enriched. There is a reference to 30 patients but there are no demographic and clinical data presented. May be not the point of the study but gender and age differences in imaging accuracy maybe interrelated.

In the discussion the future implications of this study should be addressed more and especially the points regarding sample sizes, deep learning and need for bigger datasets for the construction of a more comprehensive data model.

The prognostic use of rapid morphological measurements should also be addressed and adequately referenced.

The conclusions are fair and supported from the findings.

Reviewer 3 Report

This study reports modelling of TEVAR placement following Type B aortic dissection, for the purpose of following changes in the stent placement with time (over several years). Accurate TEVAR placement and subsequent movement are very important for the prediction of complications over time from TEVAR procedures. The data presented are compelling that this new computerised method of analysis is highly accurate and objective and is able to rapidly supply clinicians with essential clinical parameters in relation to stent placement and movement that are highly relevant to patient outcomes.

A few minor general observations:

1. The authors correctly identify as a limitation to their current work the selection of the anatomical points to provide the reference frame for their 3D coordinates. Firstly, I was a bit unclear as to where this point was exactly? It seems to be the superior, posterior, midline point of the body of thoracic vertebrae 4? Is this correct? Perhaps the authors could specify this more precisely. Secondly, the authors could speculate on other potential reference points to include in their furture work, eg the manubriosternal junction.

2. Another possible expansion of this work would be to use 4 dimensional MR images to correlate aortic flow/turbulance with stent morphology, and changes in stent morphology. The authors may wish to comment on this.

Finally, I detected a few minor typographical errors that the authors may wish to correct:

1. Line 35: "deflection angle, which IS highly complex..."

2. Line 49: "blood supply to THE aorta, it is currently..."

3. Line 257 and line 261: The authors use the term coarctation, when I think they mean dissection?

4. Figure 8: The word "method" is spelt incorrectly on each of the axes.

Reviewer 4 Report

I enjoyed reviewing this interesting and educational manuscript; however, there are some writing/editing issues that the authors should consider and address.  The following are suggestions/comments regarding these issues.  Line 32, "...method does not exceed 8 mm with an average ...".  Line 34, "...angle, which is highly complex and cannot....".  Line 49, "blood supply to the aorta, it is ...".  Lines 51 & 52, "...blood transfusion [5-7].  There have been ....".  Line 53, "...by stent implantation, having a mortality rate as high...".  Lines 62 & 63, "...found that the aortic arch presenting a more Gothic arch architecture was associated with reduced ...".  Line 65, "...were associated with a higher risk of ...".  Lines 68 & 69, "...mark degree was less likely to form a complete ...".  Linea 72 & 73, "...but error-prone, the measurement results ...".  Line 79, "...length and angles.  It is impossible to ...".  Lines 87 & 88, "...on the stent and determined their spatial coordinates and complete the calculation of the ...".  Line 104, "...are determined on the stent.  The peak points of ...".  Line 115, "...we determined the midpoint of the support ring.  For a support ...".  Line 122, "...we performed the visualization of stent parameters.  The matplotlib ...".  Line 130, "...we proposed and analyzed the following three ...".  Lines 138 & 139, "...modeling processes, the coordinates of P01, P02, and P03, and the midpoint of the stent end-loop ring were obtained for the ...".  Lines 152 & 153, "...we defined the characteristic length ...".  Line 170, "...obtained by overlapping the calibration of the ...".  Line 172, "We used the vector ...".  Lines 188 & 189, "...in excellent agreement.  The difference between ...".  Line 203, "medium complexity.  The correction of the measurement ...".  Line 218, "...deflection angles for a patient during ...".  Line 219, "...stent ring for a patient during the first ...".  Lines 256 & 257, "...after TEVER for aortic type B ...".  Line 276, "...observation point, which causes the differences between ...".  Line 296, "In this study, we used marker points ...".  Line 329, "...not only provides measurement support ...".  Line 330, "...complexity, but also makes it possible to build ...".